# Mimicking Urinary Tract Infections Caused by Uropathogenic *Escherichia coli* Using a Human Three-Dimensional Tissue Engineering Model

**DOI:** 10.3390/microorganisms12112155

**Published:** 2024-10-26

**Authors:** Félix-Antoine Pellerin, Élodie Dufresne, Stéphane Chabaud, Hazem Orabi, Stéphane Bolduc

**Affiliations:** 1Centre de Recherche en Organogénèse Expérimentale/LOEX, Regenerative Medicine Division, CHU de Québec-Université Laval Research Center, Quebec, QC G1J 1Z4, Canada; felix-antoine.pellerin@crchudequebec.ulaval.ca (F.-A.P.); stephane.chabaud@crchudequebec.ulaval.ca (S.C.);; 2Department of Urology, Assiut University, Assiut 71515, Egypt; 3Department of Surgery, Faculty of Medicine, Laval University, Quebec, QC G1V 0A6, Canada

**Keywords:** urinary tract infection, *Escherichia coli*, tissue engineering, urothelium, chitosan

## Abstract

Uropathogenic *Escherichia coli* are the main causal agent of urinary tract infections. These diseases can affect more than half of women during their lifetime. Moreover, recurrent urinary tract infections can affect up to 30% of patients, leading to higher social and economic costs for the community. No efficient treatment against the recurrent form of the disease has been discovered. Due to the low average rate of successful translation from 2D cell culture and in vivo animal models into clinical trials, new models that mimic pathologies, such as those produced by tissue engineering, are needed. A model of human-derived 3D bladder mucosa was produced by tissue engineering techniques using collagen gels and organ-specific primary human stromal and epithelial cell populations. This model was used to mimic the different steps of a urinary tract infection: adhesion, invasion, intracellular bacterial community and quiescent intracellular reservoir formation and, finally, bacteria resurgence after umbrella cell exfoliation through chitosan exposure to mimic the recurrent infection. The uropathogenic strain UTI-89-GFP was used as infectious bacteria and BL-21-GFP strain as a control. Our model is unique and is the first step toward mimicking the different phases of a UTI in a human context.

## 1. Introduction

Urinary tract infection (UTI) occurs in 60% of women during their lifetime, but also in men over 50 who become more susceptible to the onset of such urinary disorders [1]. Uropathogenic *Escherichia coli* (UPEC) will be isolated in more than 85% of cases [1]. Of patients with UTI, 20–30% will experience recurrences (rUTIs) [2], which can lead to bladder and kidney damage [3]. In the USA, it represents one million consultations per year [4]. Some women may have more than six rUTIs per year. It degrades patients’ quality of life and is associated with social and financial costs for the community, estimated to be USD 3.5 billion/year in the USA [5]. While potential reinfection by bacteria from the intestinal microbiota is possible, it has been demonstrated by molecular and clinical studies that rUTI is mainly due to the persistence of UPEC in a dormancy state inside the bladder urothelial cells (UCs) [6]. Current treatments only target UPEC outside UCs, without affecting the intracellular dormant UPEC [7]. Moreover, these antibiotic treatments could result in alterations to the microflora of the vagina and gastrointestinal tract and the selection of multidrug-resistant organisms [8,9,10]. Therefore, new strategies are needed to prevent UTIs and especially rUTIs [11].

The terminally differentiated superficial cells of the bladder urothelium (umbrella cells) are lined with the asymmetric unit membrane (AUM), which is composed of a complex of uroplakins (UPs, UPIa, Ib, II, IIIa) [12,13]. This highly specialized membrane provides the permeability barrier function of the urothelium [14]. Most UPEC strains encode filamentous, adhesive surface appendages called type 1 pili. Type 1 pili are tipped with the adhesin FimH [15], which recognizes oligomannosylated UPIa [16]. When FimH binds to UP, UPEC is internalized by a zippering mechanism consisting of a plasma membrane sheath that engulfs the bacterium [17].

A schematical summary of the infection steps is presented in Figure 1. Once intracellular, UPEC rapidly grows and divides, forming biofilm-like clusters of bacteria, termed “intracellular bacterial communities” (IBCs) [18,19,20]. The UP plaque covering the IBCs protects the UPEC by preventing neutrophils from gaining access to the bacteria [21]. Eventually, the UPECs burst into the lumen of the bladder, a phenomenon called “fluxing” [7], and the IBC cycle repeats as the escaped bacteria bind and invade the umbrella cells again (Figure 1) [18,19,20,22]. The effects of this cascade could allow UPEC to evade the host immune response and thus persist in the urinary tract. Moreover, in vitro and in vivo, it has been shown that the formation of IBCs confers protection to UPEC from being killed by antibiotics [7,23].

During infection, umbrella cells can be exfoliated, resulting in a wounded urothelial surface allowing UPEC to invade the underlying intermediate urothelial cells [24]. A dense network of actin fibers (F-actin), limiting bacterial proliferation, would trap bacteria [25]. In this situation, UPEC establishes quiescent intracellular reservoirs (QIRs), where bacteria can persist in the bladder epithelium for extended periods, protected from the immune cells and antimicrobial treatments [7,23,24,26,27]. When intermediate cells containing QIR have to differentiate in umbrella cells, during the normal urothelial turnover or in the case of damage, UPEC can be reactivated. Bacteria could grow and form IBC, which could release bacteria in the bladder lumen and induce an rUTI [24,25]. This step can be mimicked by treating the urothelium with chitosan, a derivative of chitin [28,29].

Models of UTI have been established [30], especially in mice [31], and gave us the basis of present knowledge [32]. Nevertheless, animal models are limited in mimicking the human response to drug testing. They should be replaced by innovative models such as those produced by tissue engineering [33]. We designed a tissue-engineered human-derived organ-specific 3D bladder mucosa substitute (BMS) [34,35,36]. This model is very close to the native tissue and can be helpful to better understand the steps of infection of the urothelium by UPEC and help to develop new therapeutic agents inhibiting the crucial stages of the infection, adhesion, formation of IBC and QIR and finally recurrence. As this 3D model is generated using organ-specific human cells and presents near-native bladder characteristics [37], it would allow for better translation from in vitro findings to in vivo reality, especially by limiting unwanted side effects and complications for patients. Mimicking the different phases of the infection will be the first step.

## 2. Materials and Methods

### 2.1. Ethics Statement

The current study was conducted according to the Declaration of Helsinki. It was approved by the institution’s committee for the protection of human participants (Comité d’éthique de la recherche du CHU de Québec-Université Laval, protocol code 2012-1341). All patients provided informed written consent before biopsies.

### 2.2. Bacterial Strains and Culture

UPEC strain used in this study was human cystitis strain *E. coli* UTI89 modified to express a green fluorescent protein (vsfGFP-9): SLC-719 [38]. UPEC was grown under agitation (37 °C, 150 rpm) in Luria Bertani (LB) broth (BD Difco, Franklin Lakes, NJ, USA). The non-pathological strain of *E. coli* used in this study was SLC-637, a BL21 carrying the plasmid allowing expression of fluorescent protein vsfGFP-9. Bacterial cultures were performed with or without agitation for both strains (UTI89 UPEC and BL21 control). At the indicated time, 1 ml of the bacterial culture was sampled and put in a plastic (acrylic) cuvette (Sarstedt, Nümbrecht, Germany) and optical density was measured at 600 nm using a spectrophotometer Spectromax (Molecular Devices, Sunnyvale, CA, USA).

### 2.3. Cells

To remove potential contaminants, bladder biopsies were washed in phosphate-buffered saline (PBS) containing 100 U/mL penicillin (Sigma, Oakville, ON, Canada), 25 mg/mL gentamicin (Schering, Pointe-Claire, QC, Canada) and 0.5 mg/mL Fungizone (Bristol-Myers Squibb, Montreal, QC, Canada). The samples were cut into small pieces (1 cm long, 2 mm wide) and incubated at 4 °C overnight with 10 mL of a solution containing 500 mg/mL thermolysin (Sigma) in HEPES buffer with 1 mM CaCl_2_ at pH 7.4 to allow for digestion of the basal lamina.

To isolate the bladder fibroblasts (BFs), the stroma (lamina propria) was manually separated from the urothelium, minced and incubated for 3 h at 37 °C with agitation in a solution of 0.125 U/mL collagenase H (Roche Diagnostics Canada, Montreal, QC, Canada) diluted in Dulbecco–Vogt modification of Eagle’s medium (DMEM, Invitrogen, Burlington, ON, Canada) containing 10% fetal bovine serum (Hyclone, Logan, UT, USA), 100 U/mL penicillin and 25 mg/mL gentamicin (Fb medium). After collagen digestion, the BFs were harvested by centrifugation and seeded in a culture flask at 6 × 10^4^ cells/cm^2^ in Fb medium. They were cultivated at 37 °C in a humidified 8% CO_2_ atmosphere. The medium was exchanged three times a week.

To isolate the urothelial cells (UCs), the epithelium, previously separated from the stroma, was incubated for 30 min in trypsin (Intergen Company, Purchase, NY, USA) resulting in urothelium dissociation in individual UCs. The UCs were seeded at a density of 5.333 × 10^3^/cm^2^. A feeder layer of irradiated (6000 rad) human neonate foreskin dermal fibroblasts were seeded at a density of 5.333 × 10^3^/cm^2^ one week before. UCs were cultivated in a 3:1 mix of DMEM and Ham’s F12 (Flow Lab., Mississauga, ON, Canada) supplemented with 5% FBS (GE Healthcare, Chicago, IL, USA), 24.3 μg/mL adenine (Sigma, Oakville, ON, Canada), 5 μg/mL crystallized bovine insulin (Sigma Aldrich, St. Louis, MO, USA), 1.1 μM hydrocortisone (Teva Canada Ltd., Scarborough, ON, Canada), 0.212 μg/mL isoproterenol hydrochloride (Sandoz Canada, Boucherville, QC, Canada), 10 ng/mL epidermal growth factor (Austral Biologicals, San Ramon, CA, USA) and antibiotics: 100 U/mL penicillin and 25 mg/mL gentamicin (Sigma-Aldrich) (UC medium). They were cultivated at 37 °C in a humidified 8% CO_2_ atmosphere. The medium was exchanged three times a week.

This experiment used one cell population of BFs and one of UCs. Both cell populations were incubated at 37 °C in a humidified 8% CO_2_ atmosphere. The medium was changed three times a week. Passages were performed when cells reached 80–90% confluence using trypsin. UCs were used between passages 1 to 3 and BFs between passages 2 to 4.

### 2.4. Three-Dimensional Tissue-Engineered Bladder Mucosa Substitute (Figure 2)

A modified version of the self-assembly tissue engineering protocol based on previous publications was used for these experiments [39,40]. A suspension of 6,000,000 BFs (or other density during optimization) in 3 mL Fb medium was mixed with 3 mL of 5 mg/mL PureCol-EZ collagen solution (Advanced Biomatrix, Carlsbad, CA, USA) and distributed in 6 wells of a 12-well plate (including a paper anchor) to produce 1 cm^3^ of cellularized collagen gel. The next day, 100,000 UCs were seeded on the collagen gel. Constructs were cultivated for seven days in UC medium supplemented with 50µ/mL ascorbate (Sigma, Canada). Engineered tissues were then put at the air–liquid interface using a specific device and cultivated in a UC medium for another 21 days with ascorbate.

**Figure 2 microorganisms-12-02155-f002:**
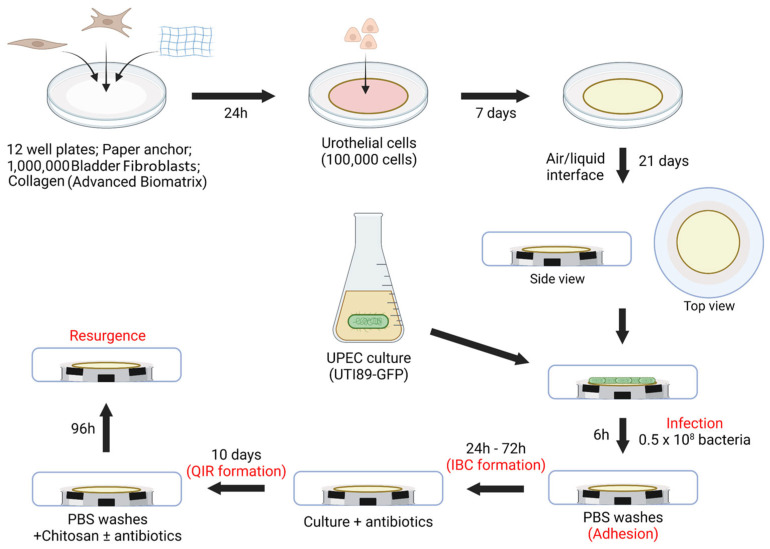
In vitro urinary tract infection protocol. Mimicking UPEC (UTI89) urinary tract infection using a 3D human-derived bladder mucosa substitute: each step mimicked is written in red.

### 2.5. Histology

Bladder mucosa substitute sections were fixed in Histochoice tissue fixative (Amresco, Solon, OH, USA) before being embedded in paraffin. Histological sections were cut 5 µm thick before being stained using Masson’s trichrome (MT). Sections were analyzed under transmitted-light brightfield with microscope Axio Imager M2 microscope (Carl Zeiss, Oberkochen, Germany).

### 2.6. Evaluation of Mannose Coverage

After bladder mucosa production using different UC seeding densities, reconstructed tissues were 1 h incubated with 10 µg/mL of the biotinylated form of lectin concanavalin A (Sigma) to allow for binding to mannosyl residues at the surface of the umbrella cells (UPIa) before being rinsed three times using PBS-BSA 1%, and subsequent 1 h incubation with 5 µg/mL streptavidin Alexa Fluor 488 (ThermoFisher Scientific, Waltham, MA, USA) was carried out. Tissues were then rinsed thrice in PBS-BSA 1% and analyzed using a Typhoon Trio+ fluorescence scanner (GE Healthcare, Chicago, IL, USA). Six tissues were imaged for each of the three conditions. The control (i.e., without streptavidin) was only carried out once.

### 2.7. Infection (Figure 2)

The last change of cell culture medium of the bladder mucosa substitutes before the infection was one using UC medium containing 10 mg/mL kanamycin instead of penicillin and gentamicin. The cultures were then transferred to an isolated chamber to avoid contamination of the environment by pathogenic bacteria. Two days before the experiment, bacteria were defrosted and cultivated on an LB agar plate supplemented with kanamycin. An isolated colony was picked and incubated in LB broth containing kanamycin for 16 h at 37 °C under agitation (200 rpm). The next day, the optical density of the culture was determined, and a new bacterial culture was started at OD = 0.1. After about three hours, bacteria were harvested, and the pellet was diluted at 10^8^ bacteria/mL in a UC medium containing kanamycin.

Bladder mucosa substitutes were transferred in a new sterile 12-well plate and incubated with 0.5 mL of diluted bacterial culture for 6 h. The tissue was then rinsed three times with PBS and incubated for 1 h with UC medium containing penicillin and gentamicin to kill any extracellular bacteria. Tissues were repositioned at the air–liquid interface for the subsequent 72 h culture with UC medium containing kanamycin. After that, UC medium containing penicillin/gentamicin was used for two weeks to kill any extracellular bacteria simulating a medical treatment of the infection.

After two weeks of antibiotics treatments, tissues were 20 min incubated in 1 ml 0.01% chitosan in UC medium with or without penicillin/gentamicin. Tissues were then rinsed and incubated for 96 h in UC medium with or without penicillin/gentamicin.

### 2.8. Scanning Electron Microscopy

Samples were harvested after the first 6 h of infection and fixed with 2.5% glutaraldehyde in 0.1 M cacodylate buffer (pH 7.4) at 4 °C, rinsed with cacodylate buffer and postfixed in 1% osmium tetroxide. The biopsies were dehydrated and then critical-point-dried. Tissue sections of each bladder mucosa substitute were spattered with gold and viewed with a Jeol JSM-63060LV microscope (Tokyo, Japan).

### 2.9. Immunofluorescences

Samples were cut from reconstructed tissues using 4 mm punch biopsy before being fixed in cold acetone, then rinsed in PBS and incubated 1 h with PBS/BSA 1% before 24 h incubation with a primary antibody AE1/AE3 diluted 1/200 and a mix of secondary antibodies Alexa Fluor 594 –red-(Abcam, Toronto, ON, Canada) diluted 1/400 with Hoechst 33,258 diluted 1/200 (Sigma, Canada). The bacteria, being GFP-positive, can be seen in green. Between each step, tissues were washed three times with PBS and finally covered with mounting medium and coverslips. The slides were observed with a Zeiss LSM 800 laser-scanning confocal microscopy system (Zeiss, North York, ON, Canada).

## 3. Results

### 3.1. Establishment of the Growth Curve for UPEC UTI-89-GFP and Non-Pathogenic Control BL21-GFP Strains

As the literature mentions, cultivating UPEC without agitation increased the expression of the type 1 pilus [41,42]. UPEC UTI-89-GFP and non-pathogenic control BL21-GFP strains were grown in LB broth in the presence of kanamycin with or without agitation. Both conditions allowed the two strains to grow with exponential and stationary phases (Figure 3 upper panels). The bacterial culture without agitation resulted in a lower optical density, indicating a lower bacteria concentration probably due to the limited access to oxygen and nutrients. A 3 h culture corresponded to the exponential phase for both strains in the two conditions. Even if the growth of the two bacterial strains does not seem different without ascorbate (0 µg/mL), with a physiological concentration (10 µg/mL) or with the concentration used for the reconstruction of BMS by tissue engineering (50 µg/mL), it is nevertheless reduced in the DME medium if we compare it with bacterial growth with the optimal medium (LB).

### 3.2. Histological Characterization of the Bladder Mucosa Substitute (BMS) Reconstructed by Tissue Engineering

The proposed 3D model is a hybrid of the completely biological human BMS using the self-assembly method and differentiation at the air–liquid interface and the collagen-gel-based method, where the differentiation occurs in submerged conditions but in the presence of retinoic acid. Both methods use organ-specific cells. Here, the stroma was a collagen gel populated by BFs and seeded with UCs. The UCs were differentiated at the air–liquid interface. As seen in Figure 4, the histology of the BMS was very close to native tissue [43], with basal lamina separating the stroma and the epithelium, a basal layer of small cuboidal cells and three to five layers of intermediate UCs showing the expected racket morphology. Finally, a well-differentiated flat UC layer comprising umbrella cells was present on the top of the reconstructed tissue. The urothelium seemed more organized when 1 million BFs were used in the stroma compared to 0.5 million. This cell density was used for the rest of the experiment (Figure 4 upper panels).

Reconstructed tissues were kept in the culture for at least 18 weeks (Figure 4 lower panels) to allow for simulating the different phases of UTI. This duration includes two complete cycles of urothelium regeneration [43]. This unique feature of the reconstructed bladder mucosa using the self-assembly technique, probably due to the preservation of stem/progenitor cells [44], makes it possible to consider long-term treatments as it could be performed with rodent animal models.

### 3.3. Characterization of the Urothelium Maturation Using Mannose Presence as a Marker

As the histological appearance of the model was very encouraging, mannosyl residue at the surface of the urothelium was investigated. Exposition at the luminal surface of the urothelium of the mannosyl residues associated with uroplakin Ia is a marker of urothelial differentiation. The mannosyl residues are also crucial for the UPEC infection of umbrella cells. It is the keyhole for entrance into these cells, whereas the FimH, present on the UPEC, is the key. The dark signal corresponds to the presence of the sugar residue. Mannosyl residues were detected by associating with a plant lectin, concanavalin A (Figure 5A). All the BMSs showed the presence of mannosyl residue, and it increased with the seeding density of UCs (Figure 5B). The black circles at the periphery correspond to the paper anchors, which retain non-specific signals.

### 3.4. Adhesion of UPEC on the Urothelial Cells

Contrary to what was expected, the condition where UPECs were grown with agitation gave a more significant GFP signal than the greater number of UTI-89-GFPs attached to urothelial cells (Figure 6A); the condition where UTI-89-GFPs were grown without agitation gave signals similar to what was obtained for the non-pathogenic strain BL21-GFP. The condition with agitation during bacterial culture was retained for the rest of the experiments. Electron microscopy allowed us to observe cells at the surface of the urothelium, whereas the uroplakin plaque, with potentially mannosyl residue, was visible (Figure 6B).

### 3.5. Invasion of Umbrella Cell Step with the Formation of IBC

The UPEC is efficiently attached to the urothelial cells, probably through the mannosyl residues present at the luminal surface of the epithelium. Pursuing the culture of the BMSs for 72 h in the presence of attached UPEC or control bacteria resulted in the presence of a GFP-rich structure inside UCs for UPEC (Figure 7). Indeed, the yellow signal results from the colocalization of the green signal (GFP) and red signal (AE1/AE3). The bacteria found in the control culture seemed to be at the surface of the cells, probably resulting from unattached or poorly attached bacteria remaining after the washing. The yellow structure corresponds to what is identified in the literature as an intracellular bacterial community (IBC).

### 3.6. Persistence of UTI89-GFP After Two Weeks of Antibiotic Treatment

After the formation of an IBC, the culture of the BMS was pursued for two weeks but in the presence of penicillin and gentamicin. Both bacterial strains, UTI-89-GFP and BL21-GFP, are resistant to kanamycin due to the plasmid they carry but remain sensitive to penicillin and gentamicin. The use of such antibiotics results in the elimination of extracellular bacteria. Only bacteria inside UCs can survive [45]. As shown in Figure 8 (2D and 3D views), in the UPEC-infected BMS, a green signal can be detected after two weeks of culture in the presence of antibiotics. The green signal (GFP) can be attributed to the presence of UTI-89-GFP persistent in the UCs after forming what is known as a quiescent intracellular reservoir (QIR). Nevertheless, it was an infrequent event in the actual model.

### 3.7. Forced Resurgence of UTI89-GFP from QIR After Chitosan Treatment Can Be Inhibited by Antibiotic Treatment

To verify that the presence of the detected signal corresponds to live bacteria and to mimic a recurrent infection [46], the BMS, after two weeks of culture in the presence of antibiotics, was treated with a chitosan solution that induced the exfoliation of the umbrella cells and induced the differentiation of intermediate cells containing the green signal (i.e., QIR). This forced resurgence of UTI-89-GFP was carried out in the presence or the absence of antibiotics. The left panel of Figure 9 shows clearly that umbrella cells were lost after chitosan treatment. The right panels of Figure 9 show that, in the absence of antibiotics, an rUTI took place, with a massive green signal indicating UPEC being present. In contrast, a weak signal could be seen in the presence of antibiotics, suggesting that the infection was limited or controlled as in [47].

## 4. Discussion

Urinary tract infections, especially their recurrent forms, are a considerable burden for health systems and patients, decreasing their quality of life. Several models have been designed using monolayer cell culture or laboratory animal models, mainly mice [11]. Recently, a few 3D models, such as organoids, have been developed [30].

Here, we proposed a human-derived 3D organ-specific model mimicking all the phases of a UTI (Figure 6, Figure 7 and Figure 8) and rUTI (Figure 9). Indeed, in this study, the BMS showed a differentiated urothelium (Figure 4) exposing at its surface mannosylated UPIa (Figure 5), which is essential for UPEC entry in urothelial cells. It showed the potential to allow for the adhesion of UPEC UTI-89-GFP on umbrella cells (Figure 6) but not the bacteria of the non-pathogenic strain BL21-GFP. These UPECs can invade urothelial superficial cells and form IBCs (Figure 7). They replicate and invade intermediate cells to form QIRs (Figure 8). These QIRs can be reactivated as they do in rUTIs to generate a productive infection (Figure 9).

Compared to monolayer models, it is clear that the three-dimensional organization of the urothelium and the presence of molecules linked to its maturation level allow BMSs to understand further the cellular and molecular mechanisms involved in UTIs and rUTIs. In particular, it is known that cancer cells, for example, express UPs in an uncoordinated manner, which leads to a failure in impermeability [48,49,50]. Moreover, the phenotype of cells in a monolayer culture often bears little relation to what is found in vivo. Animals also have significant limitations in mimicking the behavior of human tissues [11]. For example, mice, which are widely used as models, have a lower number of epithelial layers different from what is found in humans. The frequency of urination and the composition of urine are also significantly different.

While this model can undoubtedly be helpful for a better understanding of the mechanism of UTIs, it nevertheless has several limitations that must be underlined. First of all, the model is devoid of immune components. The immune system’s role in controlling and resolving UTIs is known and should be investigated. Similar tissue-engineered models have recently been designed to have an immunocompetent feature. First, a vaginal mucosa model, used to mimic HIV infection, included macrophages [51]. In this model, the macrophages were heterologous and derived from monocytes, and can be added directly in the model or added on the top of the epithelium. Second, a model of bilayered skin (dermis and epidermis), including T-cells [52], has been reconstructed to mimic psoriasis, with the objective to study the effect of the polarization of immune cells. The T-cells were heterologous. The cells were polarized in Th-1 or Th-17 T-cells before being added to the reconstructed model. Finally, a model of bilayered skin with autologous immune components (macrophages, lymphocytes and dendritic cells) was reconstructed and maintained over time [53]. In this case, the immune cells were harvested during the skin cell extraction process and banked. The choice of future immunocompetent model to use will be guided by the question to answer.

The second limitation of the model is the absence of a stretch of the bladder urothelium, which is also known to play a crucial role during UTI. Using a similar model of the bladder mucosa, a bioreactor has been designed and tested and can be used to increase the complexity of the model if needed [54].

Another limitation of the model is the source of the cells used to reconstruct the BMS. Generally, female and older patients are the most affected people. Because the cells used came from young males and only the UTI-89 strain was used, the results are not generalizable. Currently, ethical reasons limit access to the cell from more diverse patients. It could be desirable to reconstruct a BMS using cells from people more or less susceptible to developing an rUTI to understand the role of the bacterial strains and the patient’s cells. Other limitations are the use of only one strain of UPEC. Clinical isolates expressing GFP could be beneficial. In this study, a basic model was developed as proof of concept. Future recruitment needs to include women and older males without experience of UTI, with non-recurrent or recurrent experience of UTI. The bacterial strains from these patients will be isolated to be matched with the model reconstructed with the patient cells but also to create new combinations. Such models could serve as a personalized platform to test new therapeutics for people with rUTIs or to investigate the underlying mechanisms of infection and recurrences. Very small biopsies can be collected and expanded to produce the model due to cell culture conditions that allow for stem cell preservation [36]. Also, to reduce the unwanted effect of the serum in the culture on the model, especially in the case of immune component addition, a serum-free medium has been developed and will be used in the present model [55].

In the current model presented in this study, very few IBCs and QIRs can be identified in the BMS, and all the limitations described above can be responsible for that observation. Moreover, we cannot completely exclude residual antibiotics limiting the bacterial growth and formation of an IBC and QIR. Production of the model using only kanamycin should be tested. In this case, the only step using penicillin/gentamicin will be the treatment-like step. Further experiments should be conducted to try to circumvent these limitations.

Some technical points must also be highlighted. It was demonstrated that ascorbate inhibits UPEC growth. However, using ascorbate is essential to the model to allow for the maintenance of a collagen matrix over long periods. In the absence of ascorbate, the bladder fibroblasts will not be able to compensate for the tissue degradation induced by the secretion of MMP by the epithelial cells, leading to damaged tissues. Nevertheless, the growth inhibitory concentration is 1.25 mg/mL [56]. The concentration used in the model is 50 µg/mL (25-fold less), but the intracellular concentration is unknown, and the inhibitory growth concentration could vary from strain to strain. In the case of UTI-89 in our model, the ascorbate at the physiological concentration or at the used concentration did not modify the growth rate of bacteria (Figure 3 lower panels). The second technical point to discuss is using a high-glucose medium to produce the BMS. The glucose concentration in the UC medium is 3.825 g/L, when 1.1 g/L is the physiological limit for healthy people and 1.26 is the signal of potential diabetes. If some studies indicate that people affected by diabetes are more prone to UTI following changes in the glycosylation profile of Ups [57,58], it is reasonable to ask if the presence of such an amount of glucose can induce a glucose replacement of the mannose in the sugar motifs associated with UPIa. This experimental limitation can also modify the extent of the adhesion of UPECs and their invasion and subsequent steps.

Nevertheless, the model has several advantages. It is reconstructed using human cells and is organ-specific (from the bladder). These cells are primary cells and not transformed cells or cancerous cells. The 3D model replicates the histology of a normal urothelium with the appropriate number of layers and morphology of the cells. This model can be kept in culture for several weeks after it reaches its mature state (it was kept for three months in other experiments). This flexibility allows for the potential to use it as an alternative to animal models for long-term experiments. Despite very few IBCs and QIRs being detected using confocal microscopy, the chitosan treatment induced a robust reactivation of the UPEC (Figure 9), indicating that a significant number of bacteria are present in the BMS after two weeks of antibiotics treatment.

Moreover, this model was recently used to test the potential of UPEC modified in the colon by cranberry proanthocyanidins [59]. In the future, with an immunocompetent model under mechanical stimulation and using not only cells from a greater variety of patients and especially from patients with different clinical conditions but also a greater variety of bacterial strains from clinical isolates, the model will be able to go much further and make it possible to answer many still unresolved questions. This model could also be used in testing new molecules, including in the context of personalized/precision medicine, which could reduce the use of animals and give more relevant results.

## Figures and Tables

**Figure 1 microorganisms-12-02155-f001:**
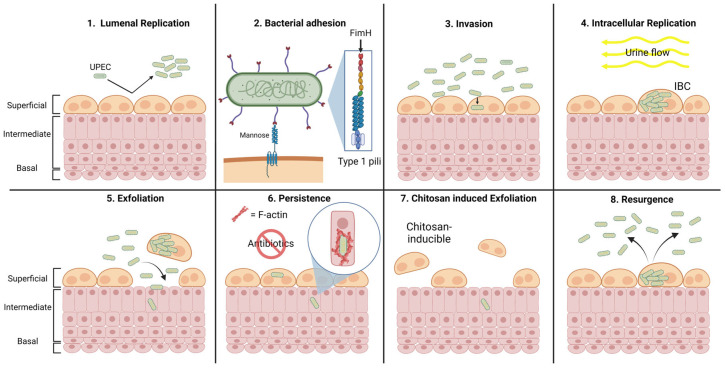
Schema of the different steps of UTI and rUTI. 1. Luminal replication, 2. Bacterial adhesion, 3. Invasion, 4. Intracellular replication, 5. Exfoliation, 6. Persistence, 7. Chitosan, Induced exfoliation, 8. Resurgence.

**Figure 3 microorganisms-12-02155-f003:**
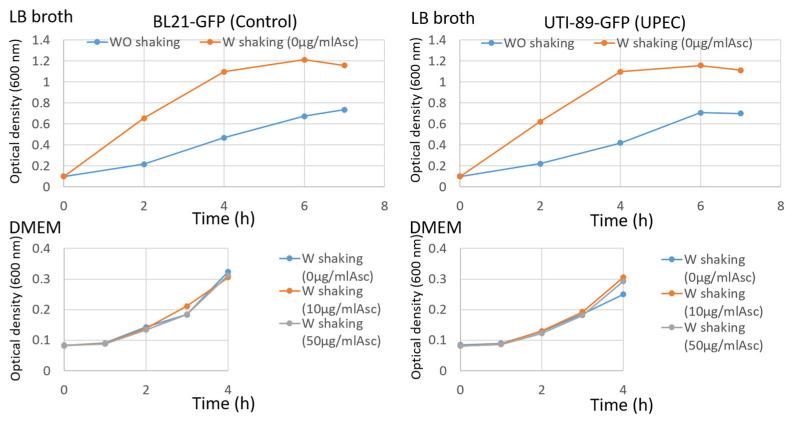
Bacterial strain growth. Bacterial strains, non-pathogenic control BL21-GFP (**left panels**) or UPEC UTI89-GFP (**right panels**), were cultivated in LB broth in the presence of kanamycin with agitation or not (**upper panels**) or in DME medium in the presence of kanamycin with agitation supplemented with 0, 10 or 50 µg/mL ascorbate (**lower panels**). The optical density of a 1 mL bacterial culture sample transferred in a plastic cuvette was measured at 600 nm using a Spectromax spectrophotometer (Molecular Devices, Kobe, Japan).

**Figure 4 microorganisms-12-02155-f004:**
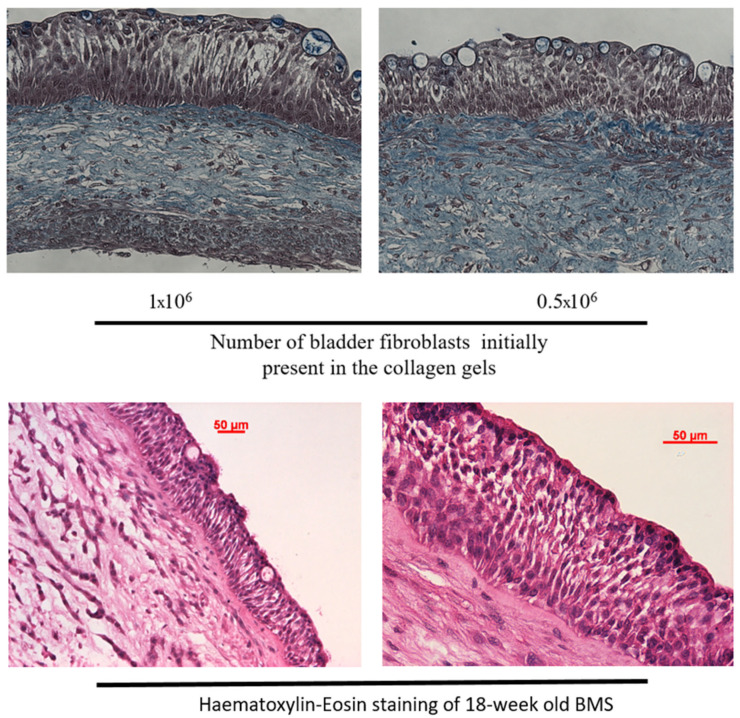
Bladder mucosa substitute histology. Twelve-well plate format BMSs were produced using the indicated number of BFs (1 × 106 or 0.5 × 106) during collagen gel casting, and 0.1 × 106 UC for the epithelial seeding step on cellularized collagen gels. Tissues were kept in culture for 3 weeks (**upper panels**) or 18 weeks (**lower panels**) at the air–liquid interface. Tissues were embedded in paraffin, sliced at 5 µm and stained using Masson’s trichrome protocol (**upper panels**) or hematoxylin–eosin (**lower panels**, 10× magnification: **right panel**, 20× magnification: **left panel**). Representative pictures are presented.

**Figure 5 microorganisms-12-02155-f005:**
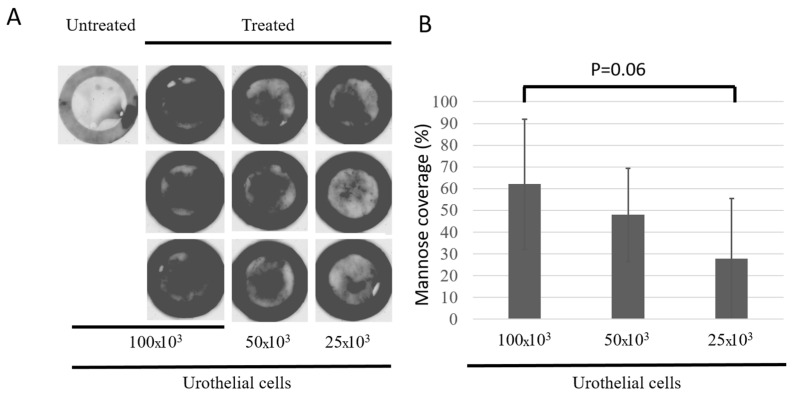
Umbrella cell mannose residue coverage. Twelve-well plate format BMSs were produced using the indicated number of UCs at the epithelial seeding step on cellularized collagen gels. (**A**). Substitutes were incubated with biotinylated concanavalin A, rinsed with PBS and incubated with streptavidin–Alexa Fluor 488 (dark color). Labeling was revealed using a Typhoon Trio+ scanner. (**B**). Area covered by mannose (i.e., glycosylated uroplakins) on BMS.

**Figure 6 microorganisms-12-02155-f006:**
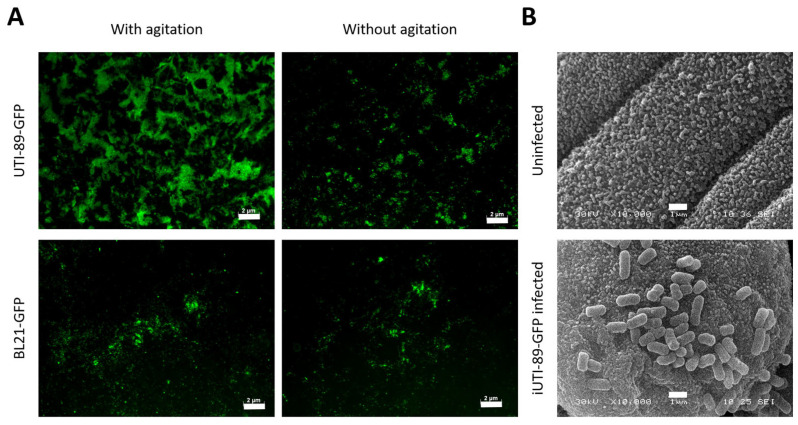
Adhesion of bacteria on urothelial cells. A. UPEC UTI-89-GFP and BL21-GFP strains were grown with or without agitation. (**A**) 1 × 10^8^ bacteria suspension was used to infect the urothelium for 1 h before the extensive wash to remove unattached bacteria, and the culture was continued for 72 additional hours. Pictures were taken to illustrate the retention of bacteria. Scalre bars are 2 µm. (**B**) Uninfected or UPEC-infected BMSs were observed by scanning electron microscopy. Scale bars are 1 µm.

**Figure 7 microorganisms-12-02155-f007:**
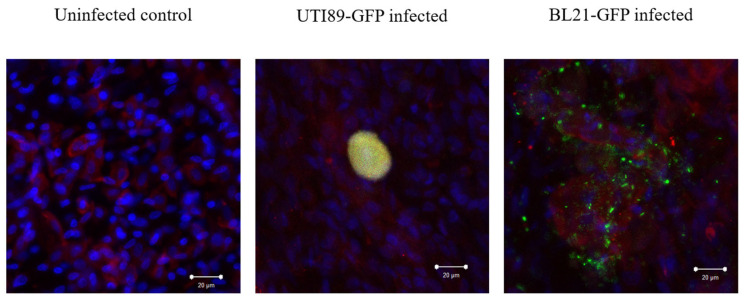
Invasion step with IBC formation. Uninfected UTI89-GFP or BL21-GFP-infected tissues were observed after a 72 h period. Green signals correspond to the GFP (bacteria), red signals to the AE1/AE3 (a cytokeratin marker specific to epithelial cells) and blue signals to Hoechst, an intercalating agent labeling the nucleus of cells. Colocalization of green and red results in a yellow signal.

**Figure 8 microorganisms-12-02155-f008:**
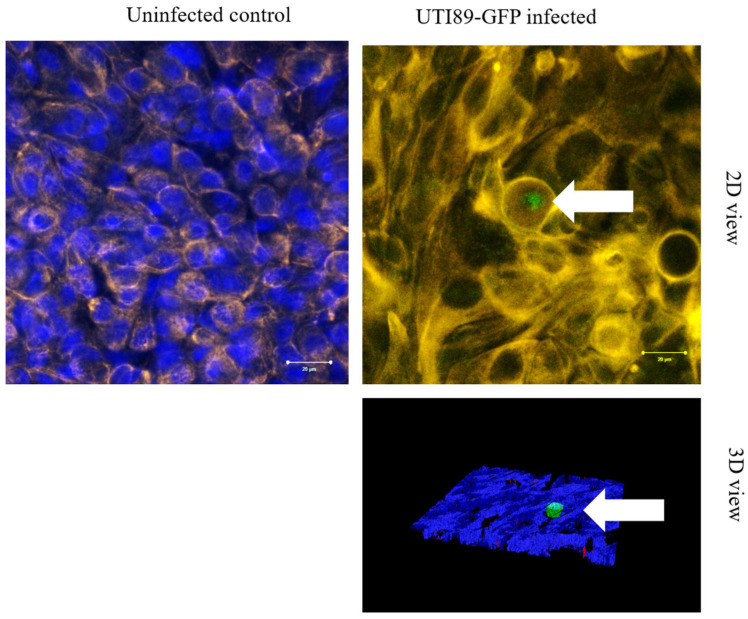
Persistence of UTI89-GFP after two weeks of antibiotic treatment. Uninfected or UTI-89-GFP-infected tissues were cultured for two weeks with antibiotics to kill extracellular bacteria. After confocal examination, the green fluorescence indicates the presence of a cluster of bacteria (arrow). Blue fluorescence is for Hoechst nuclei staining, and yellow–brown fluorescence is for AE1/AE3.

**Figure 9 microorganisms-12-02155-f009:**
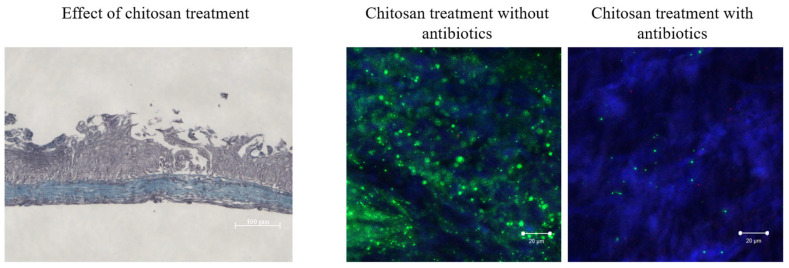
Antibiotic treatment can inhibit the forced resurgence of UTI89-GFP from QIR after chitosan treatment. UTI-89-GFP-infected BMSs kept for two weeks in culture with antibiotics were treated with chitosan to exfoliate umbrella cells (**left panel**) and force UPEC infection’s resurgence. The absence of antibiotics (**center panel**) did not allow one to control a productive disease, but the presence of the antibiotics did (**right panel**).

## Data Availability

The raw data supporting the conclusions of this article will be made available by the authors on request.

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
