# Peer review of "Mimicking Urinary Tract Infections Caused by Uropathogenic Escherichia coli Using a Human Three-Dimensional Tissue Engineering Model"

_microorganisms, 2024, doi:10.3390/microorganisms12112155_

Round 1

Reviewer 1 Report

Comments and Suggestions for Authors

The manuscript presents a valuable and innovative tissue-engineered bladder mucosa model to study urinary tract infections (UTIs), focusing on uropathogenic Escherichia coli (UPEC). The research contributes to our understanding of UTI pathogenesis, specifically in terms of bacterial adhesion and biofilm formation, while offering a model that could serve future investigations. However, there are some critical aspects of the model's design and study methodology to be clarified before being considered for acceptance.

1. The model lacks an immune system component, which is essential in the study of UTI pathogenesis, as immune cells play a critical role in bacterial clearance and infection resolution. Despite it being mentioned as one of the key limitations at the current stage, how would the model be further modified in the future?

2. The bladder cells used in the model are derived from young male donors, while UTIs primarily affect older women. This discrepancy may influence the relevance of the findings to real-world UTI cases. 

3. The study seems to use a single strain of Uropathogenic Escherichia coli (UPEC). Given the variability in pathogenicity across different strains, the findings may not be generalizable to other clinically relevant UPEC strains or other uropathogens. Tests with other/multiple clinical isolates are suggested to be added.

4. The formation of intracellular bacterial communities (IBC) and quiescent intracellular reservoirs (QIR) was mentioned as being limited in this study. What specific factors within the model might explain the low detection of IBC and QIR? 

5. The use of a high-glucose, ascorbate-containing medium could influence bacterial growth and host cell function, potentially skewing the results. I recommend conducting control experiments to assess the impact of the medium on bacterial adhesion and biofilm formation, or using alternative media more representative of bladder conditions in vivo.

6. The paper mentioned that the tissue-engineered bladder mucosa model can be kept in culture for several weeks, possibly up to three months. How do you ensure the tissue's viability and functionality over extended periods?

7. The paper does not fully explore the potential application of the model for drug testing or antibiotic resistance studies. Elaborating on how the model could be adapted for antibiotic screening or personalized therapeutic approaches would enhance the model’s clinical relevance and utility for translational research.

Author Response

Please, see the PDF file

Reviewer 2 Report

Comments and Suggestions for Authors

The paper by Pellerin et al introduces an interesting in vitro cell model to mimic the human bladder and which can be used to study the pathogenesis of bacterial, here E. coli, infection of the bladder. The data seem convincing and the model appropriate, but of course with the  limits as described in  the discussion. I would also say, that the model does not seem that easy to reproduce for other investigators, and it will  be interesting to follow whether this is possible.

Some few comments:

The authors should include in the M&M section under bacterial strains what the susceptibility (MICs) is for benzylpenicillin, gentamicin and kanamycin - this appears sporadically in  the following section. I see one problem using penicillin, that this antibiotic does actually to some degree penetrate into cells, in contrast to gentamicin. There might therefore be some antibacterial activity intracellularly when using this drug?

The authors mention in the discussion that ascorbate was as a problem for growth of the bacteria; however, i cannot find these data in  the results section ? it should be mentioned in Results before being discussed later. Ascorbate is an acid an may acidify the media - did the suthors check pH in their different media during the experiment?

Some spelling errors: E. coli is spelled as shown and should always be italized. Gentamicin is spelled with an i and not a y.

Author Response

Please, see the PDF file

Round 2

Reviewer 1 Report

Comments and Suggestions for Authors

The paper can be considered for publication after revision.